# Assessing WELBY Social Life Cycle Assessment Approach through Cobalt Mining Case Study

Anni Orola [1,*], Anna Härri [1], Jarkko Levänen [1], Ville Uusitalo [1] and Stig Irving Olsen [2]

[1] Department of Sustainability Science, Lappeenranta-Lahti University of Technology LUT, Mukkulankatu 19, 15210 Lahti, Finland
[2] Department of Technology, Management and Economics, Technical University of Denmark, Produktionstorvet, 424, 206, 2800 Kongens Lyngby, Denmark
* Correspondence: anni.orola@lut.fi; Tel.: +358-405874054

**Abstract:** The interconnected nature of social, environmental, and economic sustainability aspects must be considered in decision-making to achieve strong sustainability. Social life cycle assessment (S-LCA) has been developed to better include social sustainability aspects into life cycle thinking. However, many of the current S-LCA impact assessment approaches have been developed only on a theoretical level, and thus more case studies are needed. We assess the challenges and opportunities of the S-LCA approach through a case study on cobalt mining in the Democratic Republic of the Congo. Data for the case study were collected from scientific literature, reports, newspaper articles, and interview material. The applicability and possible strengths and weaknesses of the WELBY approach for the case were interpreted. The results showed that applying the WELBY approach in practice is possible, even though there is a lack of existing case studies. However, there are several challenges that must be addressed before the approach can be more widely used. The main challenge with the WELBY approach is the overestimation of impacts when adding multiple impact categories, as is recommended in the S-LCA guidelines. More case-specific severity weights should be developed to address this challenge. Moreover, the interpretation of the results from the perspective of informal work should be executed carefully. Even though the WELBY approach is promising, more methodological development is still needed to build a more ethical and reliable S-LCA methodology.

**Keywords:** WELBY; social life cycle assessment; S-LCA; social sustainability; Democratic Republic of the Congo

## 1. Introduction

Climate change is one of the most significant risks for humankind and the environment, thus demanding quick actions [1]. Nevertheless, the mitigation of climate impacts can cause unexpected social sustainability challenges. In sustainability science, these connections between natural and social sciences have recently been considered increasingly important [2,3]. Social sustainability can be defined as meeting all the basic needs of human life [4]. Social life cycle assessment (S-LCA) has been introduced as a new method to measure the social impacts of products or systems [5]. Together with the environmental life cycle assessment (ELCA) and life cycle cost assessment (LCC), it aims for a more holistic approach to researching sustainability impacts [6]. The new S-LCA guidelines by UNEP were released in 2020 to increase standardization in the S-LCA field [7]. However, the lack of standardization and access to suitable data are still an issue in S-LCA [5,8]. Using secondary data produced by other scientific disciplines may be challenging even when the data quality is good, whereas primary data collection may be considered too resource-consuming [7]. It is also important to note that the field of S-LCA is not uniform, and there is a lack of case studies and critical reviews considering S-LCA impact assessment methods [9].

There are two main impact assessment approaches to S-LCA: reference scale and impact pathway [7]. The reference scale approach focuses on the social performance, possible risks, and positive impacts of a product or a system [7]. The reference scale approach often utilizes databases such as the social hotspot database (SHDB) or product social impact life cycle assessment database (PSILCA) [7,10]. The impact pathway approach estimates the implications of different pathways to one endpoint, thus enabling the assessment of long-term impacts through various correlations [7,11]. For example, child labor can be assessed by using the incidence of child labor as an indicator and by calculating short-term impacts, such as health challenges due to accidents to their overall well-being, but the impacts can also be assessed through missed-education opportunities and lower-wage levels later in life [11,12].

We decided to concentrate on the impact pathway approach in our study, since it focuses more on human well-being instead of the social performance of a product system [13]. Current S-LCA impact pathway approaches are QALY, DALY, WELBY, Preston pathway, and Wilkinson pathway [7,12,14–16]. Preston pathway studies use the Preston curve to link health and wealth data together and can be used to predict the future and explain the past health effects of a population. Certain criteria, e.g., low corruption and equal wealth share among the population, must be met to use this approach [14]. As is the case with the Preston pathway, the Wilkinson pathway studies income inequality and its causal relationship to health [15]. The QALY (quality-adjusted life years) approach focuses on health impacts, and it is widely used in medical sciences [7,12]. DALY (disability-adjusted life years) is similar to QALY, and in addition to S-LCA, it is used in the Global Burden of Disease studied by the WHO [7,17]. DALY uses similar but reverse metrics as QALY, and therefore they can be utilized in QALY calculations [12]. Neither QALY nor other impact pathway approaches concentrate on well-being other than from the health perspective, and they do not cover all the indicators presented in the UNEP 2020 guidelines. Therefore, they assess social sustainability from a less comprehensive health perspective than the broader well-being assessed in WELBY (well-being-adjusted life years) [18].

The WELBY approach has been recognized as one of the most promising impact pathway approaches [19]. WELBY can be modeled as an extended QALY, as in Weidema 2006 [12,19]. WELBY has lot of similarities with QALY, but it includes other well-being impacts in addition to health, e.g., the impact of salary to human well-being [7,12]. The limitation of the original QALY approach is that it lacks sensitivity and often fails to present the worst-case scenario. Adding wellbeing metrics into the QALY approach has been estimated to solve some sensitivity-related issues [18]. Even though the WELBY approach was proposed over 15 years ago, it has been mostly developed on a theoretical level [16,18]. Only a few case studies have been executed using the QALY approach. Hannouf et al. [20] calculated the social sustainability impact of excessive working hours in Canada using the QALY approach, based on Weidema [12]. They concluded that using medical health studies as part of the S-LCA adds quality and objectivity to the research [20]. Hardadi & Pizzol [21] built their study concentrating on labor-related impacts on the same QALY approach using the EXIOBASE database. They considered using a database instead of more case-specific data one of the limitations of their study and recommended also adding more midpoint categories in future QALY studies [21].

We conducted a literature-based case study considering the S-LCA of cobalt mining in the Democratic Republic of the Congo (DRC) using the WELBY approach. Cobalt is one of the critical raw materials needed for electric vehicle batteries [22]. Transportation causes approximately 14% of the world's greenhouse gas emissions [23,24]. The electrification of traffic systems is one of the solutions to climate change mitigation [25]. The European Union has released targets to significantly reduce $CO_2$ emissions from traffic systems by 2050 [26]. The pressure to quickly expand electric vehicle (EV) production, and cobalt production with it [22], may increase the social sustainability issues caused by the mining industry. Child labor, poor health, safety of workers, stress, and violent conflicts are some of the negative social sustainability impacts related to cobalt mining in the DRC [27–36].

The EU has released the new battery regulation COM (2020)798 to address the sustainability of battery materials [37]. Even though recycling and new battery technologies aiming at reducing the need for novel cobalt in EV batteries [22,38] are under development, they might not solve the social sustainability challenges related to cobalt mining. Hence, we should find ways to measure and find solutions to these social sustainability impacts caused by the mining industry.

Some S-LCA studies concerning mine products have been conducted [39–42], but none have used the WELBY approach. Thies et al. [7,43] studied corruption, child labor, poverty, and occupational hazards in the battery value chain utilizing the reference scale approach. Bamana et al. [41] studied challenges in S-LCA data collection in the DRC cobalt-mining context. Springer et al. [42] studied the usability of S-LCA in the Brazilian artisanal and small-scale (ASM) mining context. The S-LCA has already been used in the context of cobalt mining in the DRC by Mancini et al. [44] using a more qualitative approach than the quantitative approach aimed at in this study. That study concentrated on the positive impacts of responsible sourcing initiatives on working conditions in cobalt mines.

This article focuses on assessing the WELBY approach. There is a data gap concerning practical case studies using WELBY metrics. We could not find any case studies using the WELBY approach. Assessing this approach using case studies enables recognition of practical challenges and opportunities related to it. This article focuses on answering the following questions:

- What kind of challenges and opportunities are related to the application of the WELBY approach in practice?
- What kind of research is required to strengthen the WELBY approach?

The aim of this study is to assess the WELBY approach by applying it in a case study.

## 2. Materials and Methods

### 2.1. Goal and Scope of the Case Study

The goal of this study was to apply S-LCA to quantify the social impacts of cobalt mining and identify challenges and improvement opportunities through the use of the S-LCA methodology. This comparative S-LCA study concentrated on two stakeholder groups: Congolese artisanal and small-scale mine (ASM) workers and large-scale mine (LSM) workers. ASM workers work alone or in small teams [29]. They mine cobalt in tunnels using hand tools [32]. LSM workers are employed by mining companies, and the work is done by using heavy-mining machinery [32].

The study aimed to create an impact pathway model of the social sustainability impacts caused by cobalt mining, but only short-term impacts were included due to a lack of data on long-term impacts. The WELBY approach was implemented in this study together with impact categories presented in S-LCA guidelines (Figure 1). WELBY for short-term impacts per functional unit is calculated accordingly in this study:

$$WELBY = \left( d \times n_t \times w_1 - \sum I \right) / (d \times c_a) \tag{1}$$

$$I = d \times n_e \times w_2 \tag{2}$$

where $d$ = 1 year and $w_1$ = 1:

- $d$ = duration [year];
- $n_t$ = total number of people [-];
- $n_e$ = number of people exposed to impact [-];
- $w_1$ = severity weight full health [QALY];
- $c_a$ = annual cobalt production [kg];
- $I$ = impact indicator.

The impact indicators were aggregated according to the subcategories, considering workers by S-LCA guidelines [7]. Some of the worker-related subcategories were excluded from this study. Sexual harassment was excluded, because there were no quantitative data

available. Forced labor was excluded because there are no signs of forced labor in the DRC cobalt mining [36]. There were no quantitative data concerning unequal opportunities and discrimination. Smallholders—e.g., farmers category—were recognized to be irrelevant to our study. Employment relationship was not studied as its own subcategory, even though whether a worker had an employment relationship or not was considered important. This is because the indicators that could have been connected to employment relationship, or lack of it—e.g., inadequate access to social security, or healthcare and violence related to illegal ASM—had a closer impact pathway to other subcategories.

During the inventory analysis phase, indicators were modeled by gathering data of how many people were exposed to different indicators. Following this, the indicators were multiplied using severity weights in the impact assessment phase. The final end-point result was modeled by adding all the impact indicators together and subtracting them from the full well-being of workers. The well-being of workers is the number of workers multiplied by full well-being in QALY, which is 1 [12].

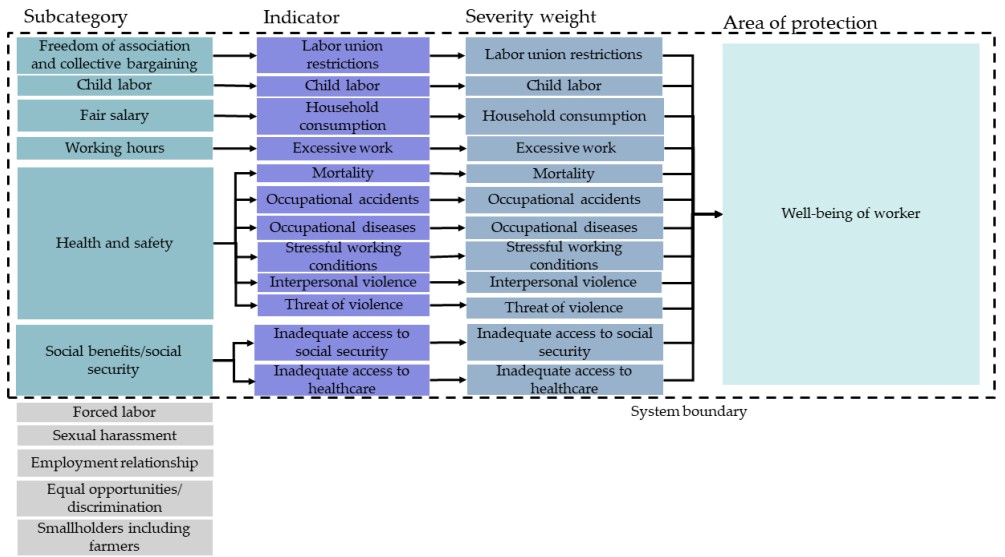

**Figure 1.** The WELBY approach and subcategories included in the case study, and the aggregation of impact indicators.

Severity weights vary between 0 and 1, where 0 means full well-being and 1 suffering as bad as death. Severity weights under 0 mean positive impacts on well-being and severity weights over 1 that the experience of well-being is considered worse than death by the person subjected to it, e.g., being tortured [12,17]. Severity weights are based on a person's subjective experiences [17]. Severity weights related to health were the same as used in the Global Burden of Disease 2019 study by WHO [17]. Severity weights for labor union restrictions, child labor, excessive work, stressful working conditions, interpersonal violence, threat of violence, inadequate access to social security, and inadequate access to healthcare are estimations by Weidema [12]. The severity weights for fair salary were based on the model by Cookson et al. [45] (Supplementary Materials). The salary-related severity weights were calculated as follows:

$$w_{i,t} = h_{i,t} + u(c_{i,t}) - 1 \tag{3}$$

$$u(c_{i,t}) = A - B \times (c_{i,t})^{1-\eta} \tag{4}$$

$$A = c_{min}^{(1-\eta)} / (c_{min}^{(1-\eta)} - c_{std}^{(1-\eta)}) \tag{5}$$

$$B = 1 / (c_{min}^{(1-\eta)} - c_{std}^{(1-\eta)}) \tag{6}$$

- $w_{i,t}$ is period-specific well-being function [QALY];
- $h_{i,t}$ is health [QALY];
- $c_{i,t}$ is consumption of individual in certain period [USD];
- $\eta$ is elasticity of the marginal value of consumption [1,26];
- $c_{std}$ is standard consumption for good standard of living [USD];
- $c_{min}$ is minimal consumption for a life worth living [USD];
- $A$ is normalization constant [-];
- $B$ is normalization constant [45].

$h_{i,t}$ is considered as 0, since the health-related social sustainability impacts are calculated separately. The severity weights and calculation data are presented in the Supplementary Materials.

The system focused on this case study was cobalt production in the DRC. The selected functional unit was the production of 1 kg of cobalt. The system boundary included the cobalt extraction and mechanical pretreatment of the mineral, e.g., washing, sorting, and crushing. Energy production, considering heavy machinery used in LSM and possible electronic tools used in ASM, was excluded from this study. The system boundary is presented in Figure 2.

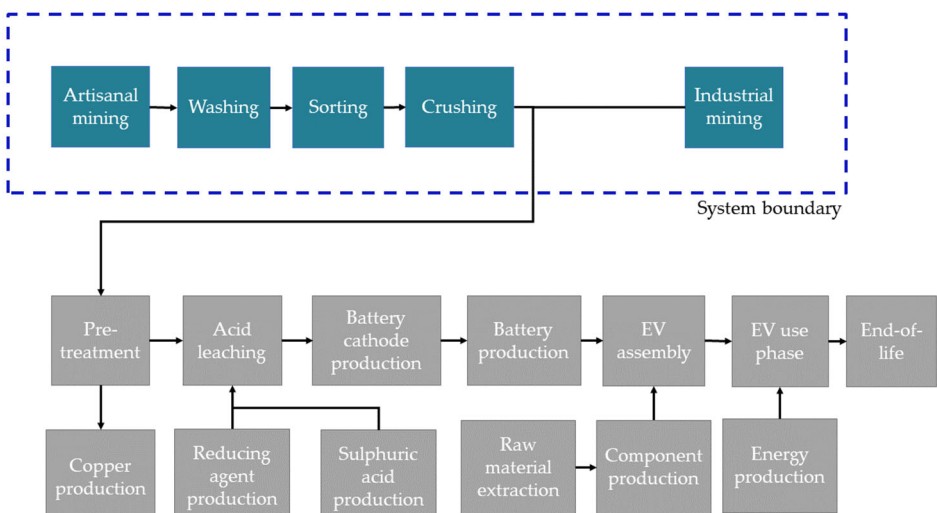

**Figure 2.** The system boundary of the case study in electric vehicle battery-manufacturing context.

The use of activity variables is recommended by UNEP [7]. The role of the activity variable is to show the relative significance of different unit processes by using worker-hours or added value [7]. In this study, the relative significance of different processes is assessed using the number of workers in different professional groups—e.g., digger and washer—instead of a time-dependent variable.

Value choices mean decisions based on ethical values [46]. It was decided that since the study only focused on the short-term impacts of cobalt mining in the DRC, it was suitable to use the life expectancy of the DRC instead of the highest possible life expectancy. One of the value choices in the QALY context is age weighting, which was not used in this study. Age weighting means giving more value to people at a certain age, e.g., the life of an adult is more valuable than a newborn baby [46]. Since the study focused only on the short-term impacts, age weighting would have been relevant only in cases of mortality. There were no data concerning the average age of workers dying in mine accidents. Therefore, it was estimated to be 25 years, which is the average age for ASM workers [47,48]. In contrast, the average age for LSM accident mortalities, 38 years, was based on the research in which mortality in different age groups was studied [49].

Part of the impacts was allocated to copper, because cobalt and copper exist in the same mineral deposits in the DRC [50]. LSM mines exploit ore that include both minerals,

whereas ASM mainly exploits ore with high cobalt concentration, but the miners may switch to mining copper due to changes in global mineral market prices [34]. Economic allocation was used in this study to allocate the impacts between cobalt and copper (Supplementary Materials). In the case of ASM, 87.8% of impacts were allocated to cobalt, and in the case of LSM, 26.3% of impacts were allocated to cobalt.

Sensitivity analysis, consistency, and completeness checks were executed. An evaluation matrix [7] was used to assess the completeness of the data.

### 2.2. Data Collection

The WELBY approach required quantitative data, which were collected from scientific papers and reports of international organizations. The human rights issues caused by cobalt mining have been an interest for multiple human rights organizations [29,51]. It was important to use artisanal and small-scale mining (ASM) data from the Lualaba and Haut-Katanga provinces as much as possible since the ASM of other minerals in the DRC is in the conflict areas, where the social sustainability impacts might be significantly different from cobalt-mining areas [41,52].

The primary data used in this study was collected by one of the co-authors for a documentary project. Pro Ethical Trade Finland conducted a field study in the DRC in May 2019. The goal of the project was to understand the human rights implications of cobalt mining in the southern DRC. The co-author spent two weeks around Lubumbashi and Kolwezi, together with a local fixer and camerawoman, to gather information and conduct interviews. Interviews were done with local NGOs ($n = 3$), a government representative, several artisanal miners ($n = 10$–$20$), communities around industrial mines ($n = 2$), and depots ($n = 2$) where artisanal miners sold their cobalt. The two NGOs were contacted beforehand due to their expertise in cobalt mining and human rights. Furthermore, multiple industrial mines were approached beforehand, but none agreed to a visit nor wanted to be interviewed. The data collection on the field was mostly done in an ad hoc way and mainly consisted of driving around the areas to find small-scale miners and other relevant actors. The aim was to obtain representation of the various actors involved in cobalt mining. The data from the interview are referred to using the term "respondent" and their respondent number (Supplementary Materials).

### 2.3. Assumptions and Limitations

Concentrating only on the short-term impacts of cobalt mining can be considered a limitation of this study. More cohort studies would be required to assess the long-term impacts of exposure to toxic minerals or the correlation between a stressful working environment and substance abuse in the ASM context. It was assumed that the DRC paid the promised social security and mining companies offered healthcare for their workers. However, there is a lot of corruption and no guarantee that things go according to legislation [53]. It should also be noted that mostly secondary data were used in this study, and several assumptions were made due to the qualitative nature of available data.

The data quality pedigree matrix was used to assess the data quality. Using the matrix in geographical and technical contexts may have some difficulties, since it is not always clear what counts as country with similar geographical conditions or industry from a similar field. For example, we could not find any data considering occupational deaths, diseases, and accidents of LSM workers in the DRC. Instead, we used data from Ghana and India. It is difficult to estimate how much gold mining in Ghana and coal mining in India differ from cobalt mining in the DRC. We chose to use data from India and Ghana since both of the areas are geographically similar to the DRC when considering physical working conditions, e.g., hot climate. The research considering whole body vibration in open-pit mining has not taken change of seasons or climate into account [54–56]. LSM cobalt mining in the DRC is open-pit mining, and therefore our main requirement was to find data considering open-pit mining since the conditions significantly differ from underground mining. As we are not experts in work ergonomics, it is difficult to assess

whether the vibration from heavy machinery caused different ergonomic problems when used in the coal mining context in India, compared to cobalt mining in the DRC. However, Howard et al. assessed that many open-pit mines have similar conditions and machinery maintenance [55]. However, in India the weekly labor hours are 48 h maximum by law [57]. This is less than the weekly working hours in the DRC cobalt-mining sector and may affect the ergonomic impacts of the mine work. It is possible that this or some other political difference makes the data from India less suitable to be used in the DRC cobalt-mining case study.

## 3. Inventory Analysis

The S-LCA was executed by creating a model using OpenLCA and literature and interview data. The results for the inventory were modeled by calculating the number of workers ($n_e$) impacted by each indicator (I) for one year (d) and then dividing it by the annual amount of cobalt ($C_a$) and copper extracted in the DRC (Equations (1) and (2)). Specific data from the DRC cobalt-mining sector were used when available, but for some indicators, LSM data from other mining sectors or countries had to be used. The results were calculated using the following values:

- ASM: 147,500 adult workers and 35,000 child workers and 15,247,500 kg cobalt;
- LSM: 30,000 workers and 72,003,000 kg cobalt.

The number of ASM workers used in this study was an average between Amnesty International [29] and Kara [58]. The number of LSM workers was acquired from an interview with respondent 1. The amount of ASM and LSM cobalt and copper used in this study is a 4-year average between 2016 and 2019 [33,34] (Supplementary Materials). The results were calculated per occupational group for the cobalt mining process. The ASM occupational groups were team leader, collector, digger, washer, carrier, and child worker. The share of different adult ASM occupational groups were modeled according to Johansson de Silva et al. 2019 [59] and child worker was modeled according to Kara 2018 [58]. The LSM workers were divided into different groups based on the salary level rather than job descriptions (Table 1). The share of different LSM occupational groups were modeled based on Radley 2020 [60]. ASM health impacts were calculated differently for diggers and non-digging worker groups [47,61]. Team leaders were considered diggers in this study.

**Table 1.** Share of different LSM occupational groups used in S-LCA model.

| Occupational Group LSM [60] | Share (%) |
|---|---|
| Hired labour (informal day labor) | 32% |
| Unskilled 1 | 21% |
| Unskilled 2 | 2.4% |
| Skilled 1 | 13% |
| Skilled 2 | 4.2% |
| Skilled 3 | 13% |
| Congolese managers | 14% |

Assessed indicators concerning workers were stressful working conditions, occupational diseases, occupational deaths, occupational accidents, freedom of association, threat of violence, interpersonal violence, inadequate access to health care, fair salary, inadequate access to social security, child labor, and excessive working hours. It was identified that lack of safety in mine work causes different health issues, e.g., accidents, physical pain, and mortality [28,29,47,61]. The noisy, hot, and unsafe working environment can cause stress for mine workers [62], and the working hours can be 10–12 h per day [29,32,59] (respondent 2). This is more than the International Labour Organization's recommendations of

48 h per week [62]. Using child labor in ASM is fairly common [29,34,58]. The salary in some ASM and LSM occupational groups is high compared to the average local income level [32,51,63]. Only workers who have contracts with industrial mining companies are offered occupational healthcare and have access to social security [32]. Clandestine mining causes violent conflicts between ASM workers and LSM companies [31]. There have been cases leading to deaths of ASM miners abused by industrial mine guards [30,32]. ASM workers and LSM day laborers do not have employment contracts with mining companies, making collective bargaining difficult [31,60]. It could be assumed that even though it is possible for ASM workers to join labor unions, their negotiation power is weak due to the partly clandestine nature of their work [31,64–66] (respondent 3). The LSM labor unions have successfully defended their rights in the past [67].

### 3.1. Stressful Working Conditions

The inventory data for stressful working conditions were assessed by modeling how many people were feeling stressed at their work. Stress impacts mental well-being and is therefore considered part of health and safety [13].

### 3.2. Occupational Diseases ASM

The number of people experiencing LSM-related occupational diseases [13]. The following medical conditions were included:

- headache;
- low back pain;
- upper limb pain;
- lower limb pain;
- skin irritation;
- hearing loss (mild, moderate, severe) [48,61].

ASM occupational groups were divided into diggers (including group leaders) and non-diggers due to different proportions of different occupational diseases [61].

### 3.3. Occupational Diseases LSM

The number of people experiencing LSM-related occupational diseases [13]. Whole body vibration is typical in industrial open-pit mining and can cause pain in different parts of the body [55]. The following medical conditions were included:

- upper limb pain;
- foot pain;
- hearing loss (mild, moderate, severe);
- hip pain;
- knee pain;
- neck pain;
- lower back pain [49].

### 3.4. Occupational Deaths ASM

Occupational deaths were modeled by assessing how many people died annually in ASM cobalt-mining accidents in the DRC [13]. The average age for accidents was set to 25, based on Elenge et al. [47], and lost life years were the years between 25 and the Congolese life expectancy of 60.7 years [68].

### 3.5. Occupational Deaths LSM

The number of LSM occupational deaths [13]. The average age of workers getting into accidents was modeled to be 38 years [49].

### 3.6. Occupational Accidents ASM

The number of ASM-related occupational accidents [13]. Occupational accidents included exposure to the following injuries:

- lower limb fracture;
- upper limb fracture;
- eye injury;
- wound;
- bruise [47].

### 3.7. Occupational Accidents LSM

The number of LSM-related occupational accidents [13]. Occupational accidents included exposure to the following injuries:

- amputation;
- burn;
- fracture;
- wound;
- bruise [56].

### 3.8. Excessive Working Hours

The number of workers working over 48 h per week [13].

### 3.9. Violence and Threat of Violence

The number of workers that had experienced violence in their work. Since the number was high in the case of the ASM workers, and conflicts between ASM workers and LSM companies are common, an assumption was made that all the ASM workers had suffered some kind of anxiety due to the risk of being subject to violence [13]. This is seen as a result of the lack of an employment relationship, since LSM companies are the ones buying the cobalt from the ASM workers [29].

### 3.10. Inadequate Access to Healthcare

The number of workers that did not have access to occupational healthcare due to not having an official employment relationship with the mining company was modeled [13].

### 3.11. Fair Salary

The number of workers in different occupational groups, e.g., washer [13].

### 3.12. Freedom of Association

The number of workers that did not have the possibility to impact their working conditions, e.g., through collective bargaining or strike [13].

### 3.13. Inadequate Access to Pension or Social Security

The number of workers lacking the possibility of a pension or social security [13].

### 3.14. Child Labor

The number of child workers working in ASM cobalt mining [13].

The inventory results are presented in Tables 2 and 3. The inventory data is modeled by calculating the percentage of workers experiencing different impacts or flows related to impact indicators. The inventory data is calculated per kg of cobalt.

**Table 2.** ASM inventory analysis results [workers/kg].

| ASM Flow | Carrier | Child Worker | Collector | Digger | Team Leader | Washer | % of Workers | Reference |
|---|---|---|---|---|---|---|---|---|
| ASM: carrier | $7.86 \times 10^{-4}$ | | | | | | 5.00% | [59] |
| ASM: child mineral collector | | $2.01 \times 10^{-3}$ | | | | | 100.00% | [59] |
| ASM: collector | | | $4.37 \times 10^{-4}$ | | | | 10.00% | [59] |
| ASM: digger | | | | $2.88 \times 10^{-3}$ | | | 36.00% | [59] |
| ASM: team leader | | | | | $8.07 \times 10^{-4}$ | | 9.00% | [59] |
| ASM: washer | | | | | | $3.58 \times 10^{-3}$ | 40.00% | [59] |
| Child labor | | $2.01 \times 10^{-3}$ | | | | | 100.00% | [58] |
| Digger: headache: tension-type | | | | $2.39 \times 10^{-3}$ | $6.70 \times 10^{-4}$ | | 83.00% | [61] |
| Digger: low back pain, moderate | | | | $2.21 \times 10^{-3}$ | $6.19 \times 10^{-4}$ | | 76.70% | [61] |
| Digger: musculoskeletal problems upper limb pain moderate | | | | $5.28 \times 10^{-4}$ | $1.48 \times 10^{-4}$ | | 18.30% | [61] |
| Digger: other musculoskeletal disorders severity level 1 (Lower limb pain) | | | | $7.64 \times 10^{-4}$ | $2.14 \times 10^{-4}$ | | 26.50% | [61] |
| Digger: skin irritation | | | | $4.90 \times 10^{-5}$ | $1.37 \times 10^{-5}$ | | 1.70% | [61] |
| Excessive work | $7.86 \times 10^{-4}$ | $2.01 \times 10^{-3}$ | $4.37 \times 10^{-4}$ | $2.88 \times 10^{-3}$ | $8.07 \times 10^{-4}$ | $3.58 \times 10^{-3}$ | 100.00% | [29,59] |
| Fracture of patella, tibia or fibula, or ankle: short-term, with or without treatment | $1.06 \times 10^{-5}$ | $2.7 \times 10^{-5}$ | $5.91 \times 10^{-6}$ | $3.90 \times 10^{-5}$ | $1.09 \times 10^{-5}$ | $4.84 \times 10^{-5}$ | 1.90% | [47] |
| Fracture of radius or ulna: short-term, with or without treatment | $1.96 \times 10^{-5}$ | $5.01 \times 10^{-5}$ | $1.09 \times 10^{-5}$ | $7.18 \times 10^{-5}$ | $2.01 \times 10^{-5}$ | $8.92 \times 10^{-5}$ | 3.50% | [47] |
| Hearing loss: mild | $2.06 \times 10^{-4}$ | $5.29 \times 10^{-4}$ | $1.15 \times 10^{-4}$ | $7.57 \times 10^{-4}$ | $2.12 \times 10^{-4}$ | $9.41 \times 10^{-4}$ | 26.25% | [69,70] |
| Inadequate access to health care | $7.86 \times 10^{-4}$ | $2.01 \times 10^{-3}$ | $4.37 \times 10^{-4}$ | $2.88 \times 10^{-3}$ | $8.07 \times 10^{-4}$ | $3.58 \times 10^{-3}$ | 100.00% | [71] |
| Inadequate access to pensions or social security | $7.86 \times 10^{-4}$ | $2.01 \times 10^{-3}$ | $4.37 \times 10^{-4}$ | $2.88 \times 10^{-3}$ | $8.07 \times 10^{-4}$ | $3.58 \times 10^{-3}$ | 100.00% | [71] |
| Labor union restrictions | $1.97 \times 10^{-4}$ | $5.04 \times 10^{-4}$ | $1.09 \times 10^{-4}$ | $7.21 \times 10^{-4}$ | $2.02 \times 10^{-4}$ | $8.96 \times 10^{-4}$ | 100.00% | [65] |
| Injury to eyes: short-term | $7.08 \times 10^{-4}$ | $1.81 \times 10^{-3}$ | $3.94 \times 10^{-4}$ | $2.60 \times 10^{-3}$ | $7.26 \times 10^{-4}$ | $3.23 \times 10^{-3}$ | 25.00% | [65] |
| Interpersonal or communal violence | $7.86 \times 10^{-4}$ | $2.01 \times 10^{-3}$ | $4.37 \times 10^{-4}$ | $2.88 \times 10^{-3}$ | $8.07 \times 10^{-4}$ | $3.58 \times 10^{-3}$ | 90.00% | [72] |
| Moderate hearing loss | $4.95 \times 10^{-5}$ | $1.27 \times 10^{-4}$ | $2.76 \times 10^{-5}$ | $1.82 \times 10^{-4}$ | $5.08 \times 10^{-5}$ | $2.26 \times 10^{-4}$ | 6.30% | [69,70] |
| Mortality | $1.40 \times 10^{-4}$ | $3.60 \times 10^{-4}$ | $7.81 \times 10^{-5}$ | $5.15 \times 10^{-4}$ | $1.44 \times 10^{-4}$ | $6.40 \times 10^{-4}$ | 0.50% | [36,47,48] |
| Non-Digger: headache: tension-type | $3.46 \times 10^{-4}$ | $8.87 \times 10^{-4}$ | $1.9210 \times 10^{-4}$ | | | $1.58 \times 10^{-3}$ | 44.00% | [61] |
| Non-digger: low back pain, moderate | $5.02 \times 10^{-4}$ | $1.29 \times 10^{-3}$ | $2.79 \times 10^{-4}$ | | | $2.29 \times 10^{-3}$ | 63.90% | [61] |
| Non-Digger: musculoskeletal problems upper limb pain moderate | $6.76 \times 10^{-5}$ | $1.73 \times 10^{-4}$ | $3.76 \times 10^{-5}$ | | | $3.08 \times 10^{-4}$ | 8.60% | [61] |
| Non-Digger: other musculoskeletal disorders severity level 1 (Lower limb pain) | $2.08 \times 10^{-4}$ | $5.34 \times 10^{-4}$ | $1.16 \times 10^{-4}$ | | | $9.50 \times 10^{-4}$ | 26.50% | [61] |
| Non-Digger: skin irritation | $4.48 \times 10^{-5}$ | $1.15 \times 10^{-4}$ | $2.49 \times 10^{-5}$ | | | $2.04 \times 10^{-4}$ | 5.70% | [61] |
| Open wound: short-term, with or without treatment | $2.48 \times 10^{-4}$ | $6.36 \times 10^{-4}$ | $1.38 \times 10^{-4}$ | $9.11 \times 10^{-4}$ | $2.55 \times 10^{-4}$ | $1.13 \times 10^{-3}$ | 44.4% | [47] |
| Other injuries of muscle and tendon | $2.81 \times 10^{-4}$ | $7.19 \times 10^{-4}$ | $1.56 \times 10^{-4}$ | $1.03 \times 10^{-3}$ | $2.88 \times 10^{-4}$ | $1.28 \times 10^{-3}$ | 50.2% | [47] |
| Severe hearing loss | $1.93 \times 10^{-5}$ | $4.94 \times 10^{-5}$ | $1.07 \times 10^{-5}$ | $7.07 \times 10^{-5}$ | $1.98 \times 10^{-5}$ | $8.78 \times 10^{-5}$ | 2.45% | [69,70] |
| Stressful working conditions | $3.93 \times 10^{-4}$ | $1.01 \times 10^{-3}$ | $2.19 \times 10^{-4}$ | $1.44 \times 10^{-3}$ | $4.03 \times 10^{-4}$ | $1.79 \times 10^{-3}$ | 50.00% | [70] |
| Threat of violence or other contact crimes | $7.86 \times 10^{-4}$ | $2.01 \times 10^{-3}$ | $4.37 \times 10^{-4}$ | $2.88 \times 10^{-3}$ | $8.07 \times 10^{-4}$ | $3.58 \times 10^{-3}$ | 100.00% | Assumption [31,32,72] |

**Table 3.** LSM inventory analysis results [workers/kg].

| LSM Flow | Congolese Manager | Hired Labor | Skilled 1 | Skilled 2 | Skilled 3 | Unskilled 1 | Unskilled 2 | % of Workers | Reference |
|---|---|---|---|---|---|---|---|---|---|
| Amputation of finger, thumb, or toe | $2.29 \times 10^{-8}$ | $5.06 \times 10^{-8}$ | $2.00 \times 10^{-8}$ | $6.57 \times 10^{-9}$ | $2.08 \times 10^{-8}$ | $3.35 \times 10^{-8}$ | $3.76 \times 10^{-9}$ | 7.00% | [73] |
| Burns of <20% total surface area without lower airway burns: short-term, with or without treatment | $3.23 \times 10^{-8}$ | $7.15 \times 10^{-8}$ | $2.83 \times 10^{-8}$ | $9.30 \times 10^{-9}$ | $2.94 \times 10^{-8}$ | $4.74 \times 10^{-8}$ | $5.31 \times 10^{-9}$ | 9.90% | [73] |
| Musculoskeletal problems upper limb pain moderate | $1.36 \times 10^{-05}$ | $3.00 \times 10^{-5}$ | $1.19 \times 10^{-5}$ | $3.90 \times 10^{-6}$ | $1.24 \times 10^{-5}$ | $1.99 \times 10^{-5}$ | $2.23 \times 10^{-6}$ | 85.65% | [56] |
| DRC: Congolese manager | $1.58 \times 10^{-5}$ | | | | | | | 14.00% | [60] |
| Excessive work | $1.58 \times 10^{-5}$ | $3.50 \times 10^{-5}$ | $1.39 \times 10^{-5}$ | $4.55 \times 10^{-6}$ | $1.44 \times 10^{-5}$ | $2.32 \times 10^{-5}$ | $2.60 \times 10^{-6}$ | 100.00% | Respondent 3; [32] |
| Foot pain moderate | $1.88 \times 10^{-6}$ | $4.17 \times 10^{-6}$ | $1.65 \times 10^{-6}$ | $5.42 \times 10^{-7}$ | $1.72 \times 10^{-6}$ | $2.76 \times 10^{-6}$ | $3.10 \times 10^{-7}$ | 11.90% | [56] |
| General fracture | $5.98 \times 10^{-8}$ | $1.32 \times 10^{-7}$ | $5.24 \times 10^{-8}$ | $1.72 \times 10^{-8}$ | $5.44 \times 10^{-8}$ | $8.76 \times 10^{-8}$ | $9.82 \times 10^{-9}$ | 18.30% | [73] |
| Hearing loss: mild | $5.22 \times 10^{-6}$ | $1.16 \times 10^{-5}$ | $4.58 \times 10^{-6}$ | $1.50 \times 10^{-6}$ | $4.76 \times 10^{-6}$ | $7.66 \times 10^{-6}$ | $8.59 \times 10^{-7}$ | 33.00% | [69] |
| Hip pain moderate | $2.26 \times 10^{-6}$ | $5.00 \times 10^{-6}$ | $1.98 \times 10^{-6}$ | $6.50 \times 10^{-7}$ | $2.06 \times 10^{-6}$ | $3.31 \times 10^{-6}$ | $3.72 \times 10^{-7}$ | 14.28% | [69] |
| Hired labor: inadequate access to health care | | $1.12 \times 10^{-5}$ | | | | | | 32.00% | [60] |
| Hired labor: inadequate access to pensions or social security | | $1.12 \times 10^{-5}$ | | | | | | 32.00% | [60] |
| Knee pain moderate | $6.78 \times 10^{-6}$ | $1.50 \times 10^{-5}$ | $5.95 \times 10^{-6}$ | $1.95 \times 10^{-6}$ | $6.18 \times 10^{-6}$ | $9.94 \times 10^{-6}$ | $1.12 \times 10^{-6}$ | 42.85% | [56] |
| Labor union restrictions | | $1.12 \times 10^{-5}$ | | | | | | 32.00% | [60] |
| LSM: skilled worker 1 | | | $1.39 \times 10^{-5}$ | | | | | 13.00% | [60] |
| LSM: skilled worker 2 | | | | $4.55 \times 10^{-6}$ | | | | 1.20% | [60] |
| LSM: skilled worker 3 | | | | | $1.44 \times 10^{-5}$ | | | 13.00% | [60] |
| LSM: unskilled worker 1 | | | | | | $2.32 \times 10^{-5}$ | | 21.00% | [60] |
| LSM: unskilled worker 2 | | | | | | | $2.60 \times 10^{-6}$ | 2.40% | [60] |
| LSM: hired labor | | $3.50 \times 10^{-5}$ | | | | | | 32.00% | [60] |
| Moderate hearing loss | $1.25 \times 10^{-6}$ | $2.77 \times 10^{-6}$ | $1.10 \times 10^{-6}$ | $3.61 \times 10^{-7}$ | $1.14 \times 10^{-6}$ | $1.84 \times 10^{-6}$ | $2.06 \times 10^{-7}$ | 7.92% | [69] |
| Mortality | $9.31 \times 10^{-8}$ | $2.06 \times 10^{-7}$ | $8.17 \times 10^{-8}$ | $2.68 \times 10^{-8}$ | $8.48 \times 10^{-8}$ | $1.37 \times 10^{-7}$ | $1.53 \times 10^{-8}$ | 0.071 fatal accidents/ 1,000,000 h | [73] |
| Neck pain moderate | $7.54 \times 10^{-6}$ | $1.67 \times 10^{-5}$ | $6.61 \times 10^{-6}$ | $2.17 \times 10^{-6}$ | $6.87 \times 10^{-6}$ | $1.10 \times 10^{-5}$ | $1.24 \times 10^{-6}$ | 47.61% | [56] |
| Non-digger: low back pain, moderate | $1.32 \times 10^{-5}$ | $2.92 \times 10^{-5}$ | $1.16 \times 10^{-5}$ | $3.80 \times 10^{-6}$ | $1.20 \times 10^{-5}$ | $1.93 \times 10^{-5}$ | $2.17 \times 10^{-6}$ | 83.33% | [56] |
| Open wound: short-term, with or without treatment | $9.66 \times 10^{-8}$ | $2.14 \times 10^{-7}$ | $8.47 \times 10^{-8}$ | $2.78 \times 10^{-8}$ | $8.80 \times 10^{-8}$ | $1.42 \times 10^{-7}$ | $1.59 \times 10^{-8}$ | 29.60% | [73] |
| Other injuries of muscle and tendon | $3.00 \times 10^{-8}$ | $6.65 \times 10^{-8}$ | $2.63 \times 10^{-8}$ | $8.64 \times 10^{-9}$ | $2.74 \times 10^{-8}$ | $4.40 \times 10^{-8}$ | $4.94 \times 10^{-9}$ | 9.20% | [73] |
| Severe hearing loss | $4.88 \times 10^{-7}$ | $1.08 \times 10^{-6}$ | $4.28 \times 10^{-7}$ | $1.40 \times 10^{-7}$ | $4.44 \times 10^{-7}$ | $7.15 \times 10^{-7}$ | $8.02 \times 10^{-8}$ | 3.08% | [69] |
| Stressful working conditions | $2.64 \times 10^{-5}$ | $1.50 \times 10^{-5}$ | $2.32 \times 10^{-5}$ | $7.60 \times 10^{-6}$ | $2.41 \times 10^{-5}$ | $9.91 \times 10^{-6}$ | $4.35 \times 10^{-6}$ | 42.70% | [74] |

## 4. Results

Our study aimed at assessing the challenges and opportunities of the WELBY approach by applying it in a case study. The results of the case study are presented as reduced well-being per kg of cobalt. The full well-being was modeled by dividing the number of workers by annual cobalt production. This was enough, since only short-term impacts were included in the study. The social sustainability impacts were reduced from full well-being. The well-being of the ASM worker was reduced significantly below zero. According to the assessment, the well-being of the LSM worker group was only slightly reduced. The positive impact caused by a higher-than-average salary increased the well-being of the LSM worker.

The results describing the ASM and LSM worker groups can be compared with each other since the same indicators have been studied concerning both. Figure 3 shows the subcategories that had impacts. The large difference between worker groups was due to the lack of social benefits/social security, worse salary, and health and safety issues in ASM mining.

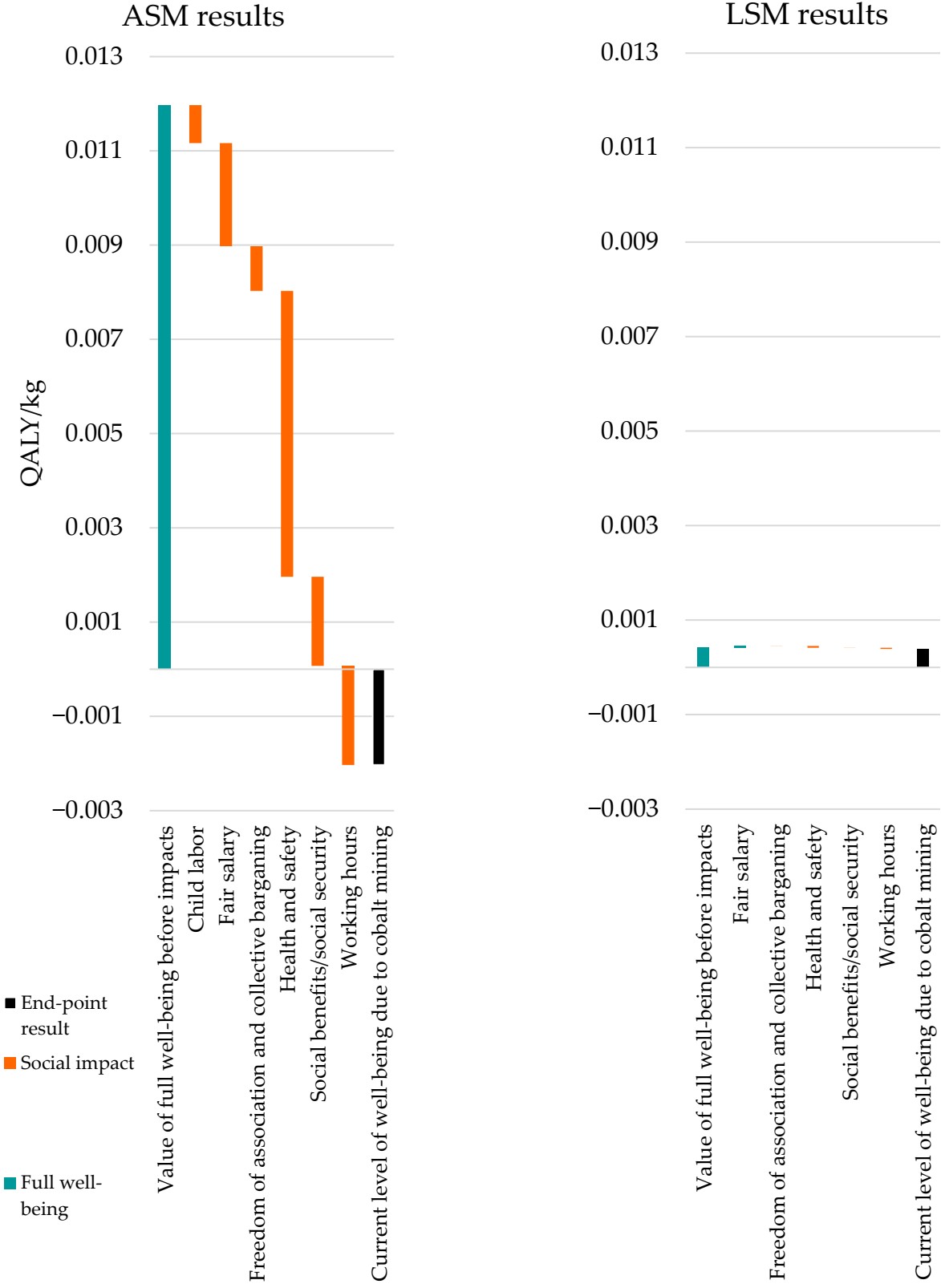

**Figure 3.** The impacts of cobalt mining on the well-being of ASM and LSM workers [QALY/kg cobalt].

The midpoint results concerning different occupational groups are presented in Figure 4 to allow deeper inspection of the results. The impacts are presented in DALYs, which is the reverse unit from QALY. The different social sustainability impacts were first modeled in DALYs and then subtracted from the full well-being to get the WELBY endpoint

result. In DALY, a unit impact of 1 means experience as bad as death and 0 life in full well-being [17], whereas in QALY it is other way round. Since the results are presented DALY/kg, the impact of 0.1 DALY/kg means that 0.1 life years have been lost due to death or disability per kg of cobalt mined in DRC. The occupational groups that had more workers had also more social sustainability impacts, since more people/kg of cobalt were exposed to the social sustainability impacts. The social sustainability impacts experienced by child workers are more visible when results are presented this way. Child workers were a relatively large part of the ASM occupational group. In addition to the negative social sustainability impacts of child labor, they also experienced all the same impacts as other non-miner worker groups. However, their salary increased their well-being to some degree. Examining the results in this form also shows that the LSM hired labor occupational group experienced more social sustainability impacts compared to other LSM occupational groups, since they were not employed long enough to achieve employment benefits such as social security.

The results of the case study showed large differences between ASM and LSM worker groups. The negative result for the ASM worker may present a major weakness in the WELBY approach. Comparison between occupational groups revealed the largest occupational groups and the occupational groups that experience the most impacts.

*Data Quality*

We assessed data quality using sensitivity analysis, and consistency and completeness checks. The data quality pedigree matrix and the results of consistency and completeness checks are presented in Appendices A–C.

Two sensitivity analyses were performed to check the possible variation in results due to uncertainties in initial data. One was to address the variation in literature, considering the number of ASM and LSM workers (Figure 5). The changes in the number of miners worked as expected, as the results are directly proportional to the number of workers. The results showed high sensitivity.

**Figure 4.** *Cont.*

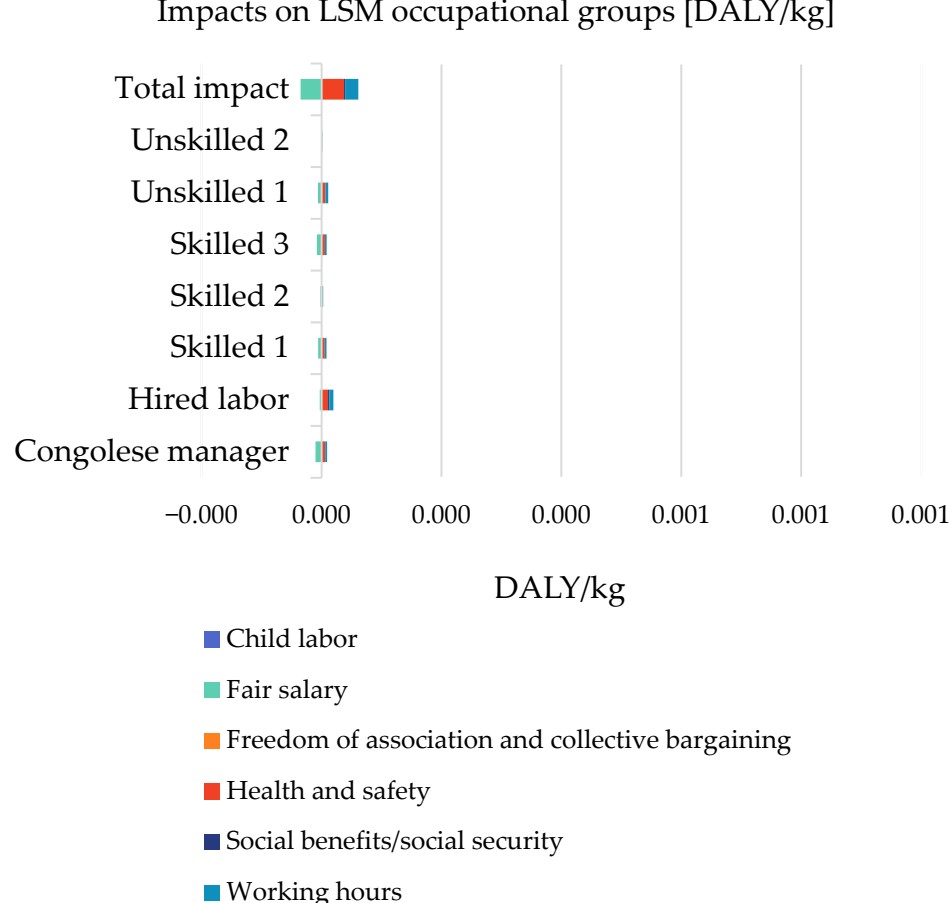

**Figure 4.** The impacts in subcategories per occupational groups [DALY/kg].

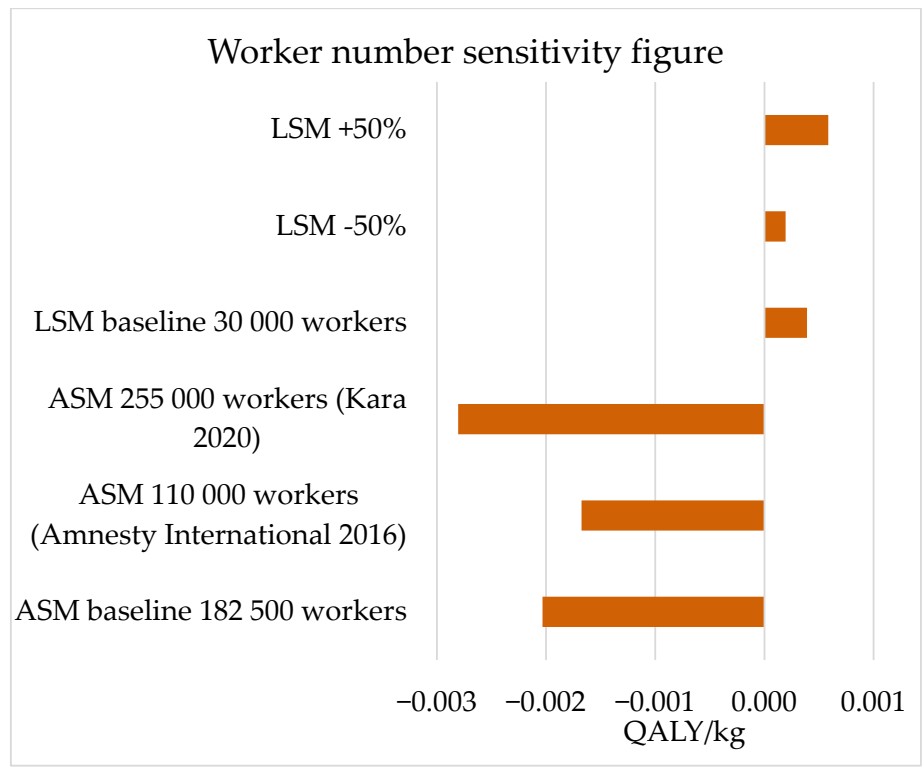

**Figure 5.** The variation in results depending on the number of workers [QALY/kg cobalt].

The second sensitivity analysis (Figure 6) concentrated on the impacts of the economic allocation between cobalt and copper. There is no data on how much copper is mined in the same sites as cobalt in all forms of cobalt mining. Therefore, the variation between different cobalt and copper grades was studied. Selected values were based on BGR 2019 and 2020 data [34] in the case of ASM mining. It was assumed that it was not economical to mine minerals below the grade of 1% [75]. The results of the sensitivity check show that there is a high variation in ASM, but low variation in LSM due to the high mass/worker ratio. High sensitivity in ASM indicates a high variety in ASM data, but a low number of uncertainties. The ASM result varying from negative to positive is due to variance in copper content in the deposit. If the mine workers mined in deposits with low cobalt and high copper content, most impacts were economically allocated to copper decreasing the well-being impacts of cobalt mining. This is caused by our scope focusing only on the impacts of cobalt mining instead of treating cobalt and copper mining as one process or system. The challenge is that there is no proper data on how much ASM workers mine copper in addition to cobalt, and how common it is to change from cobalt mining to copper mining when there are changes in the global market price. However, it is highly unlikely that ASM cobalt miners mined only or mainly 34% grade copper, because it is much higher than the average copper grade in the DRC [33].

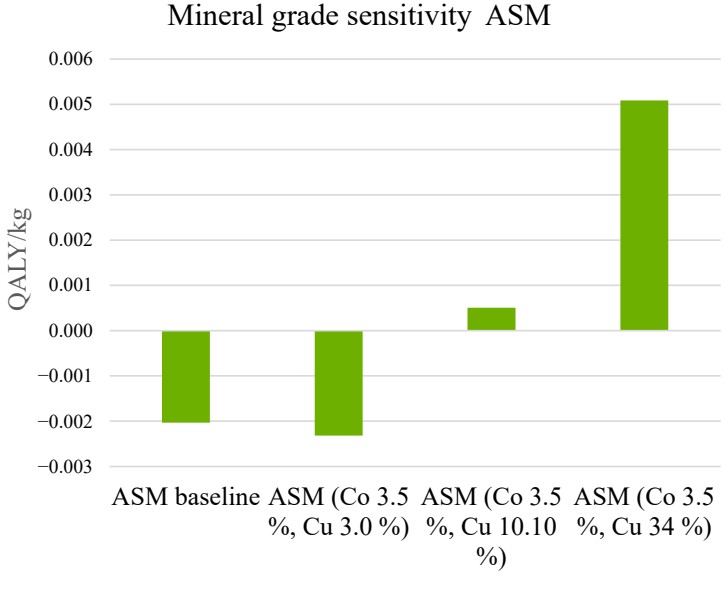

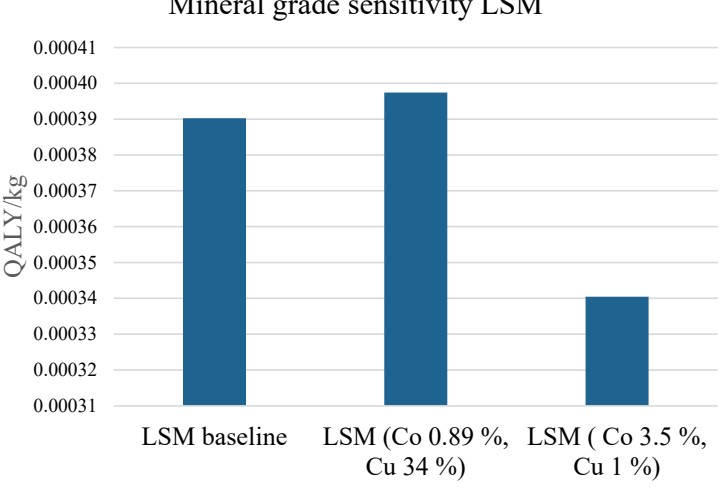

**Figure 6.** The impacts of cobalt and copper grades on variation in results due to economic allocation [QALY/kg cobalt].

## 5. Discussion

Our study aimed at researching the challenges and opportunities of the WELBY approach by applying it in a practical case study. We also assessed further research needed to improve the approach.

### 5.1. Case Study

The results of the case study show that there is a significant difference between the well-being of ASM and LSM workers. However, using more primary data would have given more context-specific results and enabled better comparison. Moreover, producing and using case-specific severity weights would have produced stronger results.

The results of our study can not be compared with the other WELBY studies, since they use different functional units than 1 kg of material. However, since our impacts are also calculated in DALYs, comparison can be performed. The results for the ASM worker group was 0.014 DALY/kg, and the result for the LSM worker group was 0.000026 DALY/kg. Arvidsson et al. modeled DALY for gold ring, where gold originated from the DRC, which was 100 DALY/kg [76]. This included DALYs from emissions, accidents, and conflicts. The number is significantly higher compared to our results since that study assumed the root cause for Congolese conflicts between 1998 and 2006 was natural resources, and they allocated 7% of millions of conflict victims to gold mining. Our study did not take Congolese armed conflicts into account, since they are mainly located in provincial regions where there is no cobalt mining [77,78]. The DALY for cobalt used in tire studs, including conflicts and accidents in the DRC mining industry, is 0.003–0.006 DALY/kg, based on Furberg et al. [40]. However, since these DALYs only concentrate on health impacts, they are not fully comparable with our study.

Mancini et al. [44] recognized many of the same social sustainability challenges concerning ASM cobalt mining assessed in this study. They found that many of these negative social impacts—e.g., mortal occupational accidents and child labor—could be reduced by responsible sourcing initiatives. However, they were not sure whether the improvements were permanent or only caused by the active piloting phase.

### 5.2. Long-Term Impacts

Our study concentrated only on the short-term impacts of cobalt mining in the DRC. Many of the occupational diseases—e.g., lung diseases—faced by miners often occur decades after the first exposure. For example, lung cancer most commonly occurs after the age of 60 [79]. Since the life expectancy at birth is relatively low, at only 60.7 years, in the DRC [68], they may not currently have a large impact on the workers' social well-being. If the currently highest life expectancy, which is 85.1 years in Hong Kong [80], was used instead, the impacts could be more severe. The highest life expectancy in African countries is 76.9 years in Algeria [81]. Part of this is due to their free universal healthcare [82]. In the DRC, there is no proper and equal healthcare system, and people must pay for their medical treatment themselves [29,83]. Due to a higher vaccination rate, Algerians have a lower child mortality rate than the DRC [84]. Whereas cobalt mining has significant negative impacts on mine workers' social well-being, it must be taken into account that improving things like the national healthcare system, in addition to working conditions, would probably also increase the well-being in mining communities.

A stressful working environment and anxiety are recognized as triggers for substance abuse. ASM miners in general are more likely to use drugs and alcohol than other professional groups in the African context, where substance abuse is generally low [85,86]. Substance abuse is also a risk factor for interpersonal violence, largely impacting women and children in ASM mining communities [87,88].

The long-term impact of child work can be severe, because children are more vulnerable to the toxic impacts of minerals [89]. In addition, not attending school affects their future earnings, since workers with higher education earn more in mines [12,63].

*5.3. Applicability of the WELBY Approach*

Developing ways to assess social sustainability is important for better understanding sustainability more holistically. S-LCA is one method for a social sustainability assessment. The main scientific contribution of this study was to assess the applicability of the WELBY approach and thus ease the path for upcoming studies using the methodology. The strength of the WELBY approach is that it measures quantitative well-being impacts more comprehensively than other S-LCA approaches [12,13]. In addition, the results are modeled at a more specific level compared to other S-LCA studies using, for example, S-LCA databases. According to Bamana et al. [41], a database-based reference scale approach does not suit countries like the DRC well, since it is a geographically large area, and the local social sustainability impacts in the mining industry differ significantly. Furthermore, the results showed high sensitivity in contrast to the QALY approach [18]. However, many challenges have to be solved before WELBY can reach its full potential.

5.3.1. Impact Pathways

As mentioned in the previous chapter, the assessment of the long-term impacts was left out from this study due to difficulties in assessment. Both data quality and the lack of existing models impacted this decision. Because of the complex dynamics of social aspects of ASM mining, it is difficult to assess the causality of multiple impact categories [42]. There are some examples of QALY impact pathway models. Hannouf et al. [20] assessed impact pathways using the Markov healthcare model. In addition, Weidema [90–92] has developed some QALY models to integrate Global Burden of Disease studies into life cycle thinking and streamline the impact pathway model using utilitarian and economic perspectives. However, none of them fit our human well-being-centered model.

We could not find any studies concentrating on the correlation between anxiety and inadequate healthcare and social security in low-income countries. The original WELBY approach assumes that everyone lacking proper access to healthcare or social security experiences anxiety or mental stress as a result [12]. Further research would be needed to better understand the connection between inadequate access to these services and mental distress from the S-LCA point of view.

5.3.2. Severity Weights

The WELBY approach was created before launching the first version of the S-LCA guidelines, and those guidelines have since been updated with new impact categories [7]. Finding matching guidelines from the literature for all impact categories of S-LCA guidelines was not possible. Therefore, it is recommended to further research and develop new severity weights matching these impact categories. This can be done using the EQ-5D questionnaire commonly used in creating QALYs for other healthcare purposes [93]. Lately, it has been used in a wider social sustainability context to create severity weights; e.g., for unemployment and different stressors at work [94,95]. The questionnaire has also been used in a comparative quality of life study considering ASM gold miners and the urban population in Zimbabwe [96]. The overall usability of the EQ-5D questionnaire in the S-LCA context could be further studied.

In addition, it would be essential to develop a severity weight for positive impacts. The social handprints—e.g., positive changes or impacts of product systems—can be created at many levels, such as the company or national level [97]. We would have liked to assess the positive impacts of LSM companies building hospitals and schools in the mining areas. The impacts of community development funds paid by the mining companies [98] could have been assessed if the scope of the study had been broader and had there been suitable severity weights. There should be a way to assess the difference between a company performing at a basic decent level and actively improving the conditions of the workers.

### 5.3.3. Interpretation of the Results

The results showed a significant enough difference in well-being of the two worker groups to make a comparison possible. Since the QALY approach has a scientific foundation, it is possible to find the most significant improvement opportunities based on these results. For example, it can be estimated that an increase of salary would also increase the well-being of the workers, or that it would be beneficial to find some kind of solution concerning the employment relationship and related social benefits of ASM workers. On the other hand, we found deeper interpretation of these results challenging. The results were aggregated on sub-categories based on UNEP guidelines [7,13]. The aggregation has been criticized before, because it makes the interpretation of the results difficult [99]. Our impression was similar: that aggregating the results this way hides at least some of the positive social sustainability impacts when the endpoint result is presented.

### 5.3.4. Ethical Challenges Caused by the Interpretation

The results for ASM workers showed negative well-being, which can be interpreted as life being worse than death. This means that workers could theoretically experience their situation in a way that they would rather be dead, which is a major ethical challenge related to this approach. It is very problematic to make these kinds of interpretations relying on secondary data without asking the opinion of the stakeholder group in concern. WELBY presents the results as reduced well-being, but it is questionable to present someone's life quality that way, especially if the result is less than 0. Even though WELBY presents the results in an easily comprehensible way, it could be better to present the results as the sum of impacts rather than reduced well-being. In this way, we would not have results that could be interpreted unethically.

It is possible that the cumulative impact of multiple categories affect well-being less than all single categories combined. At least, this is the case with DALYs. If there are multiple health issues, DALYs tend to overestimate the health impacts [17]. This is because separate severity weights are created subjectively for every medical condition [100]. Discrimination is one example of the possibility of overestimating impacts. Discrimination was ruled out of this study since there was not enough data available to assess how many people faced discrimination in cobalt mining in the DRC. Discrimination in the S-LCA context can occur due to multiple reasons, e.g., gender or race [13]. It is unclear whether discrimination should be assessed as a single indicator or with multiple different indicators. Adding multiple discrimination categories may reduce the observed well-being significantly, which may also be true, but this should be further studied to avoid counting the same impacts multiple times. Currently, WELBY could be used to estimate the impact of single impacts, but the estimation of the cumulative endpoint impact is a challenge since WELBY lacks many impact categories presented in S-LCA guidelines, and these indicators will probably add even more impacts, which makes is more likely to obtain a result of well-being reduced to below 0. Therefore, the calculation of the reduced well-being endpoint result has to be further researched and developed. Moreover, the correlation between theoretical results and actual situations should be researched, especially in cases where multiple impact categories are assessed. Comparing the results of our case study, which includes multiple impact categories, with a single indicator study created with the EQ-5D questionnaire would be interesting. Reseaching the topic by asking the mine workers to assess their own well-being subjectively would also include them in the research process. Besides, more primary data-based research is recommended until the S-LCA approach is more established.

As geographical setting may have large impact on social sustainability— for example, through changes in national legislative frameworks—it should be considered whether S-LCA can be used in comparison between countries. It can be expected, for example, that if social sustainability aspects of cobalt mining in the DRC would be compared with similar activities in Finland, Finland would probably score better. This, however, does not mean that the preferred location of mining industries should automatically be in countries such as

Finland, because it would make it even more difficult for people in lower income countries to earn their living [99,101]. This also counts for the ASM versus LSM debate, since large companies often score better because of their ability to commit to different well-being-related initiatives [102], but on the other hand, ASM work has an ability to offer income for more people compared to LSM. Our study recognized that many social sustainability impacts depend on whether the person is formally employed or not. It has been proposed that autonomy at work could be added as one of the S-LCA subcategories [42]. This could be one solution to balance the results between self-employed workers and the workers of larger companies.

In our study, the well-being impacts were assessed by modeling how the full well-being was reduced, but often the situation is more complex than this. For example, about 679–DALYs are caused annually due to protein–energy malnutrition in the DRC [103]. Thus, if these people did not work in the cobalt mining industry, they would possibly suffer from other well-being impacts due to malnutrition. Social sustainability challenges are often complex, and the consequences of making decisions—e.g., closing a cobalt mine—should also be addressed [9]. One question is whether we should even talk about well-being in the S-LCA context, since different cultures have different definitions of well-being. For example, the well-being of indigenous communities can be related to their land, and these kind of differences should be taken into account in S-LCA [104].

### 5.3.5. Link to Social Sciences

S-LCA derives from the E-LCA, which has roots in natural sciences [9]. Hence, it is interesting to compare how S-LCA relates to more traditional ways of assessing social aspects that have been utilized, especially in social sciences. It has been debated that S-LCA should take its model from the social sciences in order to make the approach scientifically stronger [105]. For example, the social impacts of mining can be studied with qualitative methods, such as semi-structured interviews and social impact assessment (SIA). Using qualitative methods enables the inclusion of data that is hard to measure numerically, such as historical, cultural and socio-cognitive factors. Even though S-LCA guidelines recommend the inclusion of these types of factors, those might be difficult to measure correctly using only the WELBY approach. This would require studying the cultural impacts on mental well-being. On the other hand, there are also other S-LCA methdologies in which the broader impacts are measured using more qualitative approaches, e.g., Likert scale in the participatory process [106,107]. These approaches have the potential to allow for the inclusion of the stakeholder groups—e.g., workers and local communities—in the reseach process, which is important to guarantee the correct interpretation of results.

### 6. Conclusions

We compared the social well-being of ASM and LSM cobalt workers in the Democratic Republic of the Congo. The well-being of both worker groups is reduced because of the mine work, but the reduction is more significant in the case of ASM workers. However, because there are no other WELBY case studies that we know of, these results are difficult to reflect.

Our study revealed the current challenges and opportunities concerning the application of the WELBY approach in a practical case study. Even though the approach seems promising and has its advantages, more research, including case studies and methodological development, is needed to build it stronger.

The negative results for ASM workers exposed how using multiple impact indicators as recommended by S-LCA guidelines causes an ethical challenge for the interpretation of the WELBY approach. When interpreting the results, the opportunities created by ASM mining become evident; e.g., employing more people compared to LSM mining.

There are not enough severity weights to cover all the S-LCA subcategories presented in the S-LCA guidelines. Since there is an issue concerning the overestimation of the impacts, it is also recommended to develop more case-specific severity weights, even

though this will add a lot of work to already-demanding data collection. Moreover, more context-specific data is needed to produce stronger results.

The development of a social life cycle assessment is crucial to measure and improve social sustainability challenges. Combined with LCA and LCC, it allows for a more holistic approach to sustainability, which is important when making decisions related to climate change mitigation in order to avoid increasing global inequality.

**Supplementary Materials:** The following are available online at https://www.mdpi.com/article/10.3390/su141811732/s1.

**Author Contributions:** Conceptualization, A.O., J.L. and V.U.; methodology, A.O.; formal analysis, A.O.; investigation, A.Hwriting—original draft preparation, A.O. and A.H.; writing—review and editing, J.L., V.U. and S.I.O.; visualization, A.O.; supervision, J.L., V.U. and S.I.O.; funding acquisition, A.O. All authors have read and agreed to the published version of the manuscript.

**Funding:** This research was funded by Maj and Tor Nessling Foundation grant number **202200201** and the APC was funded by Lappeenranta-Lahti University of Technology.

**Institutional Review Board Statement:** Ethical review and approval were waived for this study due to the reason that no primary data was collected for this study specifically.

**Informed Consent Statement:** Informed consent was obtained from all subjects involved in the study.

**Data Availability Statement:** Not applicable.

**Acknowledgments:** We would like to thank Pro Ethical Trade Finland for giving us a permission to use primary data collected for their document.

**Conflicts of Interest:** The authors declare no conflict of interest. The funders had no role in the design of the study; in the collection, analyses, or interpretation of data; in the writing of the manuscript, or in the decision to publish the results.

## Appendix A. Data Quality Matrix

**Table A1.** Data quality matrix [13].

| Scores | Reliability of the Sources | Completeness Conformance | Temporal Conformance | Geographical Conformance | Further Technical Conformance |
|---|---|---|---|---|---|
| Score 1 | Statistical study, or verified data from primary data collection from several sources. | Complete data for country-specific sector/country. | Less than 1 year difference to the time period of the dataset. | Data from same geography (country). | Data from same technology (sector). |
| Score 2 | Verified data from primary data collection from one single source, or non-verified data from primary sources, or data from recognized secondary sources. | Representative selection of country-specific sector/country. | Less than 2 years difference to the time period of the dataset. | Country with similar conditions or average of countries with slightly different conditions | Data from similar sector (e.g., within the same sector hierarchy) or average of sectors with similar technology |
| Score 3 | Non-verified data partly based on assumptions or data from non-recognized data sources. | Non-representative selection, low bias. | Less than 3 years difference to the time period of the dataset. | Average of countries with different conditions, geography under study included, with large share, or country with slightly different conditions. | Data from slightly different sector, or average of different sectors, sector under study included, with large share. |
| Score 4 | Qualified estimate (e.g., by an expert). | Non-representative selection, unknown bias. | Less than 5 years difference to the time period of the dataset. | Average of countries with different conditions, geography under study included, with small share, or not included | Average of different sectors, sector under study included, with small share, or not included. |
| Score 5 | Non-qualified estimate or unknown origin. | Single data point/completeness unknown. | Age of data unknown or data with more than 5 years difference to the time period of the dataset | Data from unknown or distinctly different regions. | Data with unknown technology/sector or from distinctly different sector. |

## Appendix B. LSM Data Quality

**Table A2.** LSM Data Quality.

| Flow | R | C | T | G | F |
|---|---|---|---|---|---|
| Amputation of finger, thumb, or toe | Score 1 | Score 2 | Score 1 | Score 2 | Score 2 |
| Burns of <20% total surface area without lower airway burns: short-term, with or without treatment | Score 1 | Score 2 | Score 1 | Score 2 | Score 2 |
| Digger: musculoskeletal problems upper limb pain moderate | Score 1 | Score 2 | Score 1 | Score 2 | Score 2 |
| DRC: Congolese manager | Score 2 | Score 2 | Score 1 | Score 1 | Score 2 |
| Excessive work | Score 1 | Score 1 | Score 1 | Score 1 | Score 1 |
| Foot pain moderate | Score 1 | Score 2 | Score 1 | Score 2 | Score 2 |
| General fracture | Score 1 | Score 2 | Score 1 | Score 2 | Score 2 |
| Hearing loss: mild | Score 1 | Score 2 | Score 1 | Score 2 | Score 2 |
| Hip pain moderate | Score 1 | Score 2 | Score 1 | Score 2 | Score 2 |
| Hired labor: inadequate access to health care | Score 1 | Score 1 | Score 1 | Score 1 | Score 2 |
| Hired labor: inadequate access to pensions or social security | Score 1 | Score 1 | Score 1 | Score 1 | Score 2 |
| Knee pain moderate | Score 1 | Score 2 | Score 1 | Score 2 | Score 2 |
| LSM: skilled worker 1 | Score 2 | Score 2 | Score 1 | Score 1 | Score 2 |
| LSM: skilled worker 2 | Score 2 | Score 2 | Score 1 | Score 1 | Score 2 |
| LSM: skilled worker 3 | Score 2 | Score 2 | Score 1 | Score 1 | Score 2 |
| LSM: unskilled worker 1 | Score 2 | Score 2 | Score 1 | Score 1 | Score 2 |
| LSM: unskilled worker 2 | Score 2 | Score 2 | Score 1 | Score 1 | Score 2 |
| LSM: hired labor | Score 2 | Score 2 | Score 1 | Score 1 | Score 2 |
| Moderate hearing loss | Score 1 | Score 2 | Score 1 | Score 2 | Score 2 |
| Mortality | Score 2 | Score 2 | Score 1 | Score 2 | Score 2 |
| Neck pain moderate | Score 1 | Score 2 | Score 1 | Score 2 | Score 2 |
| Non-digger: low back pain, moderate | Score 1 | Score 2 | Score 1 | Score 2 | Score 2 |
| Open wound: short-term, with or without treatment | Score 1 | Score 2 | Score 1 | Score 2 | Score 2 |
| Other injuries of muscle and tendon | Score 1 | Score 2 | Score 1 | Score 2 | Score 2 |
| Severe hearing loss | Score 1 | Score 2 | Score 1 | Score 2 | Score 2 |
| Stressful working conditions | Score 1 | Score 2 | Score 1 | Score 3 | Score 2 |

## Appendix C. ASM Data Quality

**Table A3.** ASM Data Quality.

| Flow | R | C | T | G | F |
|---|---|---|---|---|---|
| ASM: carrier | Score 2 | Score 2 | Score 1 | Score 1 | Score 1 |
| ASM: child mineral collector | Score 2 | Score 2 | Score 1 | Score 1 | Score 1 |
| ASM: collector | Score 2 | Score 2 | Score 1 | Score 1 | Score 1 |
| ASM: digger | Score 2 | Score 2 | Score 1 | Score 1 | Score 1 |
| ASM: team leader | Score 2 | Score 2 | Score 1 | Score 1 | Score 1 |
| ASM: washer | Score 2 | Score 2 | Score 1 | Score 1 | Score 1 |
| Child labor | Score 1 | Score 1 | Score 1 | Score 1 | Score 1 |
| Digger: headache: tension-type | Score 2 | Score 2 | Score 1 | Score 1 | Score 1 |
| Digger: low back pain, moderate | Score 2 | Score 2 | Score 1 | Score 1 | Score 1 |
| Digger: musculoskeletal problems upper limb pain moderate | Score 2 | Score 2 | Score 1 | Score 1 | Score 1 |
| Digger: other musculoskeletal disorders severity level 1 (lower limb pain) | Score 2 | Score 2 | Score 1 | Score 1 | Score 1 |
| Digger: skin irritation | Score 2 | Score 2 | Score 1 | Score 1 | Score 1 |
| Excessive work | Score 1 | Score 1 | Score 1 | Score 1 | Score 1 |
| Fracture of patella, tibia or fibula, or ankle: short-term, with or without treatment | Score 2 | Score 2 | Score 1 | Score 1 | Score 1 |
| Fracture of radius or ulna: short-term, with or without treatment | Score 2 | Score 2 | Score 1 | Score 1 | Score 1 |
| Hearing loss: mild | Score 1 | Score 1 | Score 1 | Score 1 | Score 3 |
| Inadequate access to health care | Score 1 | Score 1 | Score 1 | Score 1 | Score 1 |
| Inadequate access to pensions or social security | Score 1 | Score 1 | Score 1 | Score 1 | Score 1 |
| Injury to eyes: short-term | Score 2 | Score 2 | Score 1 | Score 1 | Score 1 |
| Interpersonal or communal violence | Score 3 | Score 3 | Score 3 | Score 1 | Score 1 |
| Labor union restrictions | Score 3 | Score 3 | Score 1 | Score 1 | Score 1 |
| Moderate hearing loss | Score 2 | Score 2 | Score 1 | Score 3 | Score 3 |
| Mortality | Score 2 | Score 2 | Score 1 | Score 1 | Score 1 |
| Non-Digger: headache: tension-type | Score 2 | Score 2 | Score 1 | Score 1 | Score 1 |
| Non-digger: low back pain, moderate | Score 2 | Score 2 | Score 1 | Score 1 | Score 1 |
| Non-Digger: musculoskeletal problems upper limb pain moderate (copy) | Score 2 | Score 2 | Score 1 | Score 1 | Score 1 |
| Non-Digger: other musculoskeletal disorders severity level 1 (lower limb pain) | Score 2 | Score 2 | Score 1 | Score 1 | Score 1 |
| Non-Digger: skin irritation | Score 2 | Score 2 | Score 1 | Score 1 | Score 1 |
| Open wound: short-term, with or without treatment | Score 2 | Score 2 | Score 1 | Score 1 | Score 1 |
| Other injuries of muscle and tendon | Score 2 | Score 2 | Score 1 | Score 1 | Score 1 |
| Severe hearing loss | Score 2 | Score 2 | Score 1 | Score 3 | Score 3 |
| Stressful working conditions | Score 1 | Score 2 | Score 1 | Score 3 | Score 2 |
| Threats of violence or other contact crimes | Score 3 | Score 3 | Score 1 | Score 1 | Score 1 |

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
