# Peer review of "Assessing WELBY Social Life Cycle Assessment Approach through Cobalt Mining Case Study"

_sustainability, doi:10.3390/su141811732_

Round 1
Reviewer 1 Report (New Reviewer)
Dear authors,
You study is interesting. Your work is valuable to further develop S-LCA, in particular when applying it in African cases. However, the methods section is lacking. You need to cite the WELBY (or WALY?) method accurately and explain how you calculate impacts. For instance, how eq. 1 is applies in the case of social benefits?
Please also improve the figure and tables captions because they need to stand on their own, i.e., be self explanatory.
You can find my extended comments in the PDF file.
success!

Author Response
Thank you for your comments! Reply as an attachment.

Reviewer 2 Report (New Reviewer)
The authors mention that "The main challenges include data availability and correct interpretation of the results". This is not clear. The interpretation of the result, by definition, depends on several aspects linked to the scope of the study, data availability and not most minor, the analyst's expertise (i.e., subjectivity). In other words, data availability and interpretation are not separate aspects as the sentence suggests. my recommendation is that the author specify better in which aspect hampers the interpretation (may it be the aggregation of Social indicators or negative WELBY results?)
L. 95-98. I recommend that the authors contextualise/include the New Battery Regulation " COM(2020)798" in this paragraph.
Equations 1 and 2. please, indicate the final unit for WELBY and I.
equation 2. it is unclear how the sum of indicators can give a unit consistent with time*person. can you please provide more details on the math?
equation 4. Shouldn't the "C" be lowercase? if not, explain what it represents
indicate the units for c_i,t A and B (even if they are dimensionless)
L. 127 "The system of the study was cobalt production in the DRC. " is unclear.
L. 237-238: "e.g., the life of 237 an adult is more valuable than a newborn baby". is this always true in any QUALY context or is specific of study [47]?
L. 255-257: "The results for the inventory were modelled by calculating the number of 255 workers impacted by each indicator for one year and then dividing it by the annual 256 amount of cobalt and copper extracted in the DRC". is there any link with the terms of the equations? is yes, please indicate which are the terms of the equations 1-2 in this sentence.
the results should be contrasted versus [42]. how the results obtained are in line/complement/contrast the results of [42]?
Author Response
Thank you for your comments! Response as attachement.

Reviewer 3 Report (New Reviewer)
This paper aims to assess the challenges and opportunities of the SLCA methodology through a case study on cobalt mining in the Democratic Republic of Congo.
According to the authors, the development of life cycle assessment is crucial to measure and improve social sustainability challenges.
SLCA, combined with LCA AND LCC, allows for a more holistic approach to analysing social sustainability; this is crucial for the mitigation of changes currently taking place, such as climate change or those related to global inequalities.
The article is not perfectly structured, abstracts and conclusions are particularly concise and lacking in key information and parameters.
The topic has been dealt with comprehensively and is of great scientific value.
The manuscript is clear, with an excellent percentage of bibliographical references from the last five years.
The manuscript is scientifically sound and the experimental design is appropriate for the verification of the design hypotheses.
Figures/schedules/tables and images show the analysed data correctly and are easy to interpret.
The conclusions are consistent with the evidence and arguments presented but are too weak.
The study is clear, complete and relevant to the field under investigation.
The objectives of the contribution, as well as the method/approach used, are well addressed and there is good logical coherence between the different parts of the paper.
The paper is discrete from a syntactic-grammatical point of view, easy to understand and pleasant to read.
All in all, the paper under review, in addition to having important bibliographical/dissemination properties, presents many points of scientific interest.
Author Response
Thank you for your comments! Response as attachment.

Round 2
Reviewer 1 Report (New Reviewer)
Dear authors,
Thank you for addressing my comments. There are a couple of minor things still left to change.
Success!

Author Response
Thank you for the comments!

This manuscript is a resubmission of an earlier submission. The following is a list of the peer review reports and author responses from that submission.
Round 1
Reviewer 1 Report
The manuscript deals with a very relevant topic and the contribution is important to move forward with assessing the social sustainability of products and services. All in all, I believe this work has the potential to be published, however, not in its current state. First, the manuscript could gain from some restructuring of certain paragraphs, some additions to the Introduction, method and results section. Throughout the manuscript I stumbled across sentences which left me wondering on how, why or what? So a lot of question on my end which means that information is missing to make this work fully transparent and reproducible.
Additionally, the contribution to science of this work is not discussed – probably because the state of research and research gap comes to short.
In the following some general remarks about this work:
Text and language: The manuscript would benefit from a professional proof read, but especially check the consistency, transitions, spelling mistakes, unprecise phrasings, comma signs in English writing and dot at the end of sentences.
Figures and tables: consider to integrate some tables (or condensed versions thereof) from the appendix into the main text and also think about putting some tables into a supplementary material – the appendix is too long and not practicable for the reader. Also the formatting of the figures in the results section should be improved (size; readability of text).
Method description: I also have some concerns in regards to how the method is described and the data used is presented – see below for more detailed information.
Introduction & background
The introduction contains now a very brief reflection on the state of research and research gap, but also a description of the case region and even some results. This is a bit confusing to the reader since the introduction should just provide information on the background, the state of research and the research gap and drawn from that the aim of this study. The state of research comes rather short at this point.
Materials and method
Also, here the structure is rather confusing: you start with data collection (SLCI) then describing the methodology but also the state of research in regards to impact pathways (so SLCIA) and a research gap – shouldn’t that be part of the introduction? Then in the goal and scope you describe the method how you calculate the impacts and severity.
Inventory
Almost no information on the indicators chosen for the subcategories nor on the data used or the respective sources is provided. Though, a reference to the appendixes is made – however, also there no explanation is provided.
Also, which indicators did you choose and which data sources did you use? Why those indicators for the subcategories? And you really couldn’t find data for discrimination?
Results
Please provide more detail on the results – contribution of subcategories; differences between ASM & LSM, … Did you also check the uncertainties connected with the inventory data? Please add the unit in the figure axis.
discussion
Nicely and clear structured and some important points are discussed.
· Which kind of negative social impacts? Could you provide some examples (the same is true in the results section) P14l422
· No data for unequal opportunities but you found evidence that this is the case – so isn’t that date for this subcategory? P14l430
· What kind of information was provided by the respondents and how? Expert interview? Shouldn’t that be explained in the method section. The same applies for all the other data used for the calculation
· What is the scientific contribution?
Reviewer 2 Report
Please see attached file.
